# Learning Disentangled Representations for Recommendation

**Jianxin Ma**[1,2]*, **Chang Zhou**[1]*, **Peng Cui**[2], **Hongxia Yang**[1], **Wenwu Zhu**[2]

[1]Alibaba Group,  [2]Tsinghua University

majx13fromthu@gmail.com, ericzhou.zc@alibaba-inc.com,
cuip@tsinghua.edu.cn, yang.yhx@alibaba-inc.com, wwzhu@tsinghua.edu.cn

## Abstract

User behavior data in recommender systems are driven by the complex interactions of many latent factors behind the users' decision making processes. The factors are highly entangled, and may range from high-level ones that govern user intentions, to low-level ones that characterize a user's preference when executing an intention. Learning representations that uncover and disentangle these latent factors can bring enhanced robustness, interpretability, and controllability. However, learning such disentangled representations from user behavior is challenging, and remains largely neglected by the existing literature. In this paper, we present the MACRo-mIcro Disentangled Variational Auto-Encoder (MacridVAE) for learning disentangled representations from user behavior. Our approach achieves macro disentanglement by inferring the high-level concepts associated with user intentions (e.g., to buy a shirt or a cellphone), while capturing the preference of a user regarding the different concepts separately. A micro-disentanglement regularizer, stemming from an information-theoretic interpretation of VAEs, then forces each dimension of the representations to independently reflect an isolated low-level factor (e.g., the size or the color of a shirt). Empirical results show that our approach can achieve substantial improvement over the state-of-the-art baselines. We further demonstrate that the learned representations are interpretable and controllable, which can potentially lead to a new paradigm for recommendation where users are given fine-grained control over targeted aspects of the recommendation lists.

## 1 Introduction

Learning representations that reflect users' preference, based chiefly on user behavior, has been a central theme of research on recommender systems. Despite their notable success, the existing user behavior-based representation learning methods, such as the recent deep approaches [49, 32, 31, 52, 11, 18], generally neglect the complex interaction among the latent factors behind the users' decision making processes. In particular, the latent factors can be highly entangled, and range from macro ones that govern the intention of a user during a session, to micro ones that describe at a granular level a user's preference when implementing a specific intention. The existing methods fail to disentangle the latent factors, and the learned representations are consequently prone to mistakenly preserve the confounding of the factors, leading to non-robustness and low interpretability.

Disentangled representation learning, which aims to learn factorized representations that uncover and disentangle the latent explanatory factors hidden in the observed data [3], has recently gained much attention. Not only can disentangled representations be more *robust*, i.e., less sensitive to the misleading correlations presented in the limited training data, the enhanced *interpretability* also finds

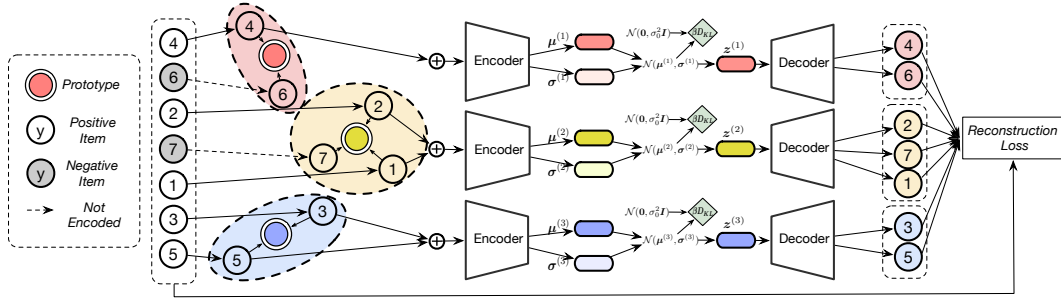

Figure 1: Our framework. Macro disentanglement is achieved by learning a set of prototypes, based on which the user intention related with each item is inferred, and then capturing the preference of a user about the different intentions separately. Micro disentanglement is achieved by magnifying the KL divergence, from which a term that penalizes total correlation can be separated, with a factor of $\beta$.

direct application in recommendation-related tasks, such as transparent advertising [33], customer-relationship management, and explainable recommendation [51, 17]. Moreover, the *controllability* exhibited by many disentangled representations [19, 14, 10, 8, 9, 25] can potentially bring a new paradigm for recommendation, by giving users explicit control over the recommendation results and providing a more interactive experience. However, the existing efforts on disentangled representation learning are mainly from the field of computer vision [28, 15, 20, 30, 53, 14, 10, 39, 19].

Learning disentangled representations based on user behavior data, a kind of discrete relational data that is fundamentally different from the well-researched image data, is challenging and largely unexplored. Specifically, it poses two challenges. First, the co-existence of macro and micro factors requires us to to separate the two levels when performing disentanglement, in a way that preserves the hierarchical relationships between an intention and the preference about the intention. Second, the observed user behavior data, e.g., user-item interactions, are discrete and sparse in nature, while the learned representations are continuous. This implies that the majority of the points in the high-dimensional representation space will not be associated with any behavior, which is especially problematic when one attempts to investigate the interpretability of an isolated dimension by varying the value of the dimension while keeping the other dimensions fixed.

In this paper, we propose the MACRo-mIcro Disentangled Variational Auto-Encoder (MacridVAE) for learning disentangled representations based on user behavior. Our approach explicitly models the separation of macro and micro factors, and performs disentanglement at each level. Macro disentanglement is achieved by identifying the high-level concepts associated with user intentions, and separately learning the preference of a user regarding the different concepts. A regularizer for micro disentanglement, derived by interpreting VAEs [27, 44] from an information-theoretic perspective, is then strengthened so as to force each individual dimension to reflect an independent micro factor. A beam-search strategy, which handles the conflict between sparse discrete observations and dense continuous representations by finding a smooth trajectory, is then proposed for investigating the interpretability of each isolated dimension. Empirical results show that our approach can achieve substantial improvement over the state-of-the-art baselines. And the learned disentangled representations are demonstrated to be interpretable and controllable.

## 2 Method

In this section, we present our approach for learning disentangled representations from user behaivor.

### 2.1 Notations and Problem Formulation

A user behavior dataset $\mathcal{D}$ consists of the interactions between $N$ users and $M$ items. The interaction between the $u^{\text{th}}$ user and the $i^{\text{th}}$ item is denoted by $x_{u,i} \in \{0, 1\}$, where $x_{u,i} = 1$ indicates that user $u$ explicitly adopts item $i$, whereas $x_{u,i} = 0$ means there is no recorded interaction between the two. For convenience, we use $\mathbf{x}_u = \{x_{u,i} : x_{u,i} = 1\}$ to represent the items adopted by user $u$. The goal is to learn user representations $\{\mathbf{z}_u\}_{u=1}^N$ that achieves both macro and micro disentanglement. We use $\boldsymbol{\theta}$ to denote the set that contains all the trainable parameters of our model.

**Macro disentanglement**  Users may have very diverse interests, and interact with items that belong to many high-level concepts, e.g., product categories. We aim to achieve macro disentanglement, by learning a factorized representation of user $u$, namely $\mathbf{z}_u = [\mathbf{z}_u^{(1)}; \mathbf{z}_u^{(2)}; \ldots; \mathbf{z}_u^{(K)}] \in \mathbb{R}^{d'}$, where $d' = Kd$, assuming that there are $K$ high-level concepts. The $k^{\text{th}}$ component $\mathbf{z}_u^{(k)} \in \mathbb{R}^d$ is for capturing the user's preference regarding the $k^{\text{th}}$ concept. Additionally, we infer a set of one-hot vectors $\mathbf{C} = \{\mathbf{c}_i\}_{i=1}^{M}$ for the items, where $\mathbf{c}_i = [c_{i,1}; c_{i,2}; \ldots; c_{i,K}]$. If item $i$ belongs to concept $k$, then $c_{i,k} = 1$ and $c_{i,k'} = 0$ for any $k' \neq k$. We jointly infer $\{\mathbf{z}_u\}_{u=1}^{N}$ and $\mathbf{C}$ unsupervisedly.

**Micro disentanglement**  High-level concepts correspond to the intentions of a user, e.g., to buy clothes or a cellphone. We are also interested in disentangling a user's preference at a more granular level regarding the various aspects of an item. For example, we would like the different dimensions of $\mathbf{z}_u^{(k)}$ to individually capture the user's preferred sizes, colors, etc., if concept $k$ is clothing.

## 2.2  Model

We start by proposing a generative model that encourages macro disentanglement. For a user $u$, our generative model assumes that the observed data are generated from the following distribution:

$$p_{\boldsymbol{\theta}}(\mathbf{x}_u) = \mathbb{E}_{p_{\boldsymbol{\theta}}(\mathbf{C})} \left[ \int p_{\boldsymbol{\theta}}\left(\mathbf{x}_u \mid \mathbf{z}_u, \mathbf{C}\right) p_{\boldsymbol{\theta}}(\mathbf{z}_u) \, d\mathbf{z}_u \right], \qquad (1)$$

$$p_{\boldsymbol{\theta}}\left(\mathbf{x}_u \mid \mathbf{z}_u, \mathbf{C}\right) = \prod_{x_{u,i} \in \mathbf{x}_u} p_{\boldsymbol{\theta}}(x_{u,i} \mid \mathbf{z}_u, \mathbf{C}). \qquad (2)$$

The meanings of $\mathbf{x}_u, \mathbf{z}_u, \mathbf{C}$ are described in the previous subsection. We have assumed $p_{\boldsymbol{\theta}}(\mathbf{z}_u) = p_{\boldsymbol{\theta}}(\mathbf{z}_u \mid \mathbf{C})$ in the first equation, i.e., $\mathbf{z}_u$ and $\mathbf{C}$ are generated by two independent sources. Note that $\mathbf{c}_i = [c_{i,1}; c_{i,2}; \ldots; c_{i,K}]$ is one-hot, since we assume that item $i$ belongs to exactly one concept. And $p_{\boldsymbol{\theta}}(x_{u,i} \mid \mathbf{z}_u, \mathbf{C}) = Z_u^{-1} \cdot \sum_{k=1}^{K} c_{i,k} \cdot g_{\boldsymbol{\theta}}^{(i)}(\mathbf{z}_u^{(k)})$ is a categorical distribution over the $M$ items, where $Z_u = \sum_{i=1}^{M} \sum_{k=1}^{K} c_{i,k} \cdot g_{\boldsymbol{\theta}}^{(i)}(\mathbf{z}_u^{(k)})$ and $g_{\boldsymbol{\theta}}^{(i)} : \mathbb{R}^d \rightarrow \mathbb{R}_+$ is a shallow neural network that estimates how much a user with a given preference is interested in item $i$. We use sampeld softmax [23] to estimate $Z_u$ based on a few sampled items when $M$ is very large.

**Macro disentanglement**  We assume above that the user representation $\mathbf{z}_u$ is sufficient for predicting how the user will interact with the items. And we further assume that using the $k^{\text{th}}$ component $\mathbf{z}_u^{(k)}$ alone is already sufficient if the prediction is about an item from concept $k$. This design explicitly encourages $\mathbf{z}_u^{(k)}$ to capture preference regarding only the $k^{\text{th}}$ concept, as long as the inferred concept assignment matrix $\mathbf{C}$ is meaningful. We will describe later the implementation details of $p_{\boldsymbol{\theta}}(\mathbf{C})$, $p_{\boldsymbol{\theta}}(\mathbf{z}_u)$ and $g_{\boldsymbol{\theta}}^{(i)}(\mathbf{z}_u^{(k)})$. Nevertheless, we note that $p_{\boldsymbol{\theta}}(\mathbf{C})$ requires careful design to prevent mode collapse, i.e., the degenerate case where almost all items are assigned to a single concept.

**Variational inference**  We follow the variational auto-encoder (VAE) paradigm [27, 44], and optimize $\boldsymbol{\theta}$ by maximizing a lower bound of $\sum_u \ln p_{\boldsymbol{\theta}}(\mathbf{x}_u)$, where $\ln p_{\boldsymbol{\theta}}(\mathbf{x}_u)$ is bounded as follows:

$$\ln p_{\boldsymbol{\theta}}(\mathbf{x}_u) \geq \mathbb{E}_{p_{\boldsymbol{\theta}}(\mathbf{C})} \left[ \mathbb{E}_{q_{\boldsymbol{\theta}}(\mathbf{z}_u \mid \mathbf{x}_u, \mathbf{C})}[\ln p_{\boldsymbol{\theta}}(\mathbf{x}_u \mid \mathbf{z}_u, \mathbf{C})] - D_{\mathrm{KL}}(q_{\boldsymbol{\theta}}(\mathbf{z}_u \mid \mathbf{x}_u, \mathbf{C}) \| p_{\boldsymbol{\theta}}(\mathbf{z}_u)) \right]. \qquad (3)$$

See the supplementary material for the derivation of the lower bound. Here we have introduced a variational distribution $q_{\boldsymbol{\theta}}(\mathbf{z}_u \mid \mathbf{x}_u, \mathbf{C})$, whose implementation also encourages macro disentanglement and will be presented later. The two expectations, i.e., $\mathbb{E}_{p_{\boldsymbol{\theta}}(\mathbf{C})}[\cdot]$ and $\mathbb{E}_{q_{\boldsymbol{\theta}}(\mathbf{z}_u \mid \mathbf{x}_u, \mathbf{C})}[\cdot]$, are intractable, and are therefore estimated using the Gumbel-Softmax trick [22, 41] and the Gaussian re-parameterization trick [27], respectively. Once the training procedure is finished, we use the mode of $p_{\boldsymbol{\theta}}(\mathbf{C})$ as $\mathbf{C}$, and the mode of $q_{\boldsymbol{\theta}}(\mathbf{z}_u \mid \mathbf{x}_u, \mathbf{C})$ as the representation of user $u$.

**Micro disentanglement**  A natural strategy to encourage micro disentanglement is to force statistical independence between the dimensions, i.e., to force $q_{\boldsymbol{\theta}}(\mathbf{z}_u^{(k)} \mid \mathbf{C}) \approx \prod_{j=1}^{d} q_{\boldsymbol{\theta}}(z_{u,j}^{(k)} \mid \mathbf{C})$, so that each dimension describes an isolated factor. Here $q_{\boldsymbol{\theta}}(\mathbf{z}_u \mid \mathbf{C}) = \int q_{\boldsymbol{\theta}}(\mathbf{z}_u \mid \mathbf{x}_u, \mathbf{C}) p_{\mathrm{data}}(\mathbf{x}_u) \, d\mathbf{x}_u$. Fortunately, the Kullback–Leibler (KL) divergence term in the lower bound above does provide a way to encourage independence. Specifically, the KL term of our model can be rewritten as:

$$\mathbb{E}_{p_{\mathrm{data}}(\mathbf{x}_u)} \left[ D_{\mathrm{KL}}(q_{\boldsymbol{\theta}}(\mathbf{z}_u \mid \mathbf{x}_u, \mathbf{C}) \| p_{\boldsymbol{\theta}}(\mathbf{z}_u)) \right] = I_q(\mathbf{x}_u; \mathbf{z}_u) + D_{\mathrm{KL}}(q_{\boldsymbol{\theta}}(\mathbf{z}_u \mid \mathbf{C}) \| p_{\boldsymbol{\theta}}(\mathbf{z}_u)). \qquad (4)$$

See the supplementary material for the proof. Similar decomposition of the KL term has been noted for the original VAEs previously [1, 25, 9]. Penalizing the latter KL term would encourage independence between the dimensions, if we choose a prior that satisfies $p_{\boldsymbol{\theta}}(\mathbf{z}_u) = \prod_{j=1}^{d'} p_{\boldsymbol{\theta}}(z_{u,j})$. On the other hand, the former term $I_q(\mathbf{x}_u; \mathbf{z}_u)$ is the mutual information between $\mathbf{x}_u$ and $\mathbf{z}_u$ under $q_{\boldsymbol{\theta}}(\mathbf{z}_u \mid \mathbf{x}_u, \mathbf{C}) \cdot p_{\text{data}}(\mathbf{x}_u)$. Penalizing $I_q(\mathbf{x}_u; \mathbf{z}_u)$ is equivalent to applying the information bottleneck principle [47, 2], which encourages $\mathbf{z}_u$ to ignore as much noise in the input as it can and to focus on merely the essential information. We therefore follow $\beta$-VAE [19], and strengthen these two regularization terms by a factor of $\beta \gg 1$, which brings us to the following training objective:

$$\mathbb{E}_{p_{\boldsymbol{\theta}}(\mathbf{C})} \left[ \mathbb{E}_{q_{\boldsymbol{\theta}}(\mathbf{z}_u \mid \mathbf{x}_u, \mathbf{C})}[\ln p_{\boldsymbol{\theta}}(\mathbf{x}_u \mid \mathbf{z}_u, \mathbf{C})] - \beta \cdot D_{\text{KL}}(q_{\boldsymbol{\theta}}(\mathbf{z}_u \mid \mathbf{x}_u, \mathbf{C}) \| p_{\boldsymbol{\theta}}(\mathbf{z}_u)) \right]. \quad (5)$$

## 2.3 Implementation

In this section, we describe the implementation of $p_{\boldsymbol{\theta}}(\mathbf{C})$, $p_{\boldsymbol{\theta}}(x_{u,i} \mid \mathbf{z}_u, \mathbf{C})$ (the decoder), $p_{\boldsymbol{\theta}}(\mathbf{z}_u)$ (the prior), $q_{\boldsymbol{\theta}}(\mathbf{z}_u \mid \mathbf{x}_u, \mathbf{C})$ (the encoder), and propose an efficient strategy to combat mode collapse. The parameters $\boldsymbol{\theta}$ of our implementation include: $K$ concept prototypes $\{\mathbf{m}_k\}_{k=1}^K \in \mathbb{R}^{K \times d}$, $M$ item representations $\{\mathbf{h}_i\}_{i=1}^M \in \mathbb{R}^{M \times d}$ used by the decoder, $M$ context representations $\{\mathbf{t}_i\}_{i=1}^M \in \mathbb{R}^{M \times d}$ used by the encoder, and the parameters of a neural network $f_{\text{nn}} : \mathbb{R}^d \to \mathbb{R}^{2d}$. We optimize $\boldsymbol{\theta}$ to maximize the training objective (see Equation 5) using Adam [26].

**Prototype-based concept assignment**    A straightforward approach would be to assume $p_{\boldsymbol{\theta}}(\mathbf{C}) = \prod_{i=1}^M p(\mathbf{c}_i)$ and parameterize each categorical distribution $p(\mathbf{c}_i)$ with its own set of $K-1$ parameters. This approach, however, would result in over-parameterization and low sample efficiency. We instead propose a prototype-based implementation. To be specific, we introduce $K$ concept prototypes $\{\mathbf{m}_k\}_{k=1}^K$ and reuse the item representations $\{\mathbf{h}_i\}_{i=1}^M$ from the decoder. We then assume $\mathbf{c}_i$ is a one-hot vector drawn from the following categorical distribution $p_{\boldsymbol{\theta}}(\mathbf{c}_i)$:

$$\mathbf{c}_i \sim \text{CATEGORICAL}\left(\text{SOFTMAX}([s_{i,1}; s_{i,2}; \ldots; s_{i,K}])\right), \quad s_{i,k} = \text{COSINE}(\mathbf{h}_i, \mathbf{m}_k)/\tau, \quad (6)$$

where $\text{COSINE}(\mathbf{a}, \mathbf{b}) = \mathbf{a}^\top \mathbf{b} / (\|\mathbf{a}\|_2 \|\mathbf{b}\|_2)$ is the cosine similarity, and $\tau$ is a hyper-parameter that scales the similarity from $[-1, 1]$ to $[-\frac{1}{\tau}, \frac{1}{\tau}]$. We set $\tau = 0.1$ to obtain a more skewed distribution.

**Preventing mode collapse**    We use cosine similarity, instead of the inner product similarity adopted by most existing deep learning methods [32, 31, 18]. This choice is crucial for preventing mode collapse. In fact, with inner product, the majority of the items are highly likely to be assigned to a single concept $\mathbf{m}_{k'}$ that has an extremely large norm, i.e., $\|\mathbf{m}_{k'}\|_2 \to \infty$, even when the items $\{\mathbf{h}_i\}_{i=1}^M$ correctly form $K$ clusters in the high-dimensional Euclidean space. And we observe empirically that this phenomenon does occur frequently with inner product (see Figure 2e). In contrast, cosine similarity avoids this degenerate case due to the normalization. Moreover, cosine similarity is related with the Euclidean distance on the unit hypersphere, and the Euclidean distance is a proper metric that is more suitable for inferring the cluster structure, compared to inner product.

**Decoder**    The decoder predicts which item out of the $M$ ones is mostly likely to be clicked by a user, when given the user's representation $\mathbf{z}_u = [\mathbf{z}_u^{(1)}; \mathbf{z}_u^{(2)}; \ldots; \mathbf{z}_u^{(K)}]$ and the one-hot concept assignments $\{\mathbf{c}_i\}_{i=1}^M$. We assume that $p_{\boldsymbol{\theta}}(x_{u,i} \mid \mathbf{z}_u, \mathbf{C}) \propto \sum_{k=1}^K c_{i,k} \cdot g_{\boldsymbol{\theta}}^{(i)}(\mathbf{z}_u^{(k)})$ is a categorical distribution over the $M$ items, and define $g_{\boldsymbol{\theta}}^{(i)}(\mathbf{z}_u^{(k)}) = \exp(\text{COSINE}(\mathbf{z}_u^{(k)}, \mathbf{h}_i)/\tau)$. This design implies that $\{\mathbf{h}_i\}_{i=1}^M$ will be micro-disentangled if $\{\mathbf{z}_u^{(k)}\}_{u=1}^N$ is micro-disentangled, as the two's dimensions are aligned.

**Prior & Encoder**    The prior $p_{\boldsymbol{\theta}}(\mathbf{z}_u)$ needs to be factorized in order to achieve micro disentanglement. We therefore set $p_{\boldsymbol{\theta}}(\mathbf{z}_u)$ to $\mathcal{N}(\mathbf{0}, \sigma_0^2 \mathbf{I})$. The encoder $q_{\boldsymbol{\theta}}(\mathbf{z}_u \mid \mathbf{x}_u, \mathbf{C})$ is for computing the representation of a user when given the user's behavior data $\mathbf{x}_u$. The encoder maintains an additional set of context representations $\{\mathbf{t}_i\}_{i=1}^M$, rather than reusing the item representations $\{\mathbf{h}_i\}_{i=1}^M$ from the decoder, which is a common practice in the literature [32]. We assume $q_{\boldsymbol{\theta}}(\mathbf{z}_u \mid \mathbf{x}_u, \mathbf{C}) = \prod_{k=1}^K q_{\boldsymbol{\theta}}(\mathbf{z}_u^{(k)} \mid \mathbf{x}_u, \mathbf{C})$, and represent each $q_{\boldsymbol{\theta}}(\mathbf{z}_u^{(k)} \mid \mathbf{x}_u, \mathbf{C})$ as a multivariate normal distribution with a diagonal covariance matrix $\mathcal{N}(\boldsymbol{\mu}_u^{(k)}, [\text{diag}(\boldsymbol{\sigma}_u^{(k)})]^2)$, where the mean and

the standard deviation are parameterized by a neural network $f_{\mathrm{nn}} : \mathbb{R}^d \to \mathbb{R}^{2d}$:

$$(\mathbf{a}_u^{(k)}, \mathbf{b}_u^{(k)}) = f_{\mathrm{nn}} \left( \frac{\sum_{i:x_{u,i}=+1} c_{i,k} \cdot \mathbf{t}_i}{\sqrt{\sum_{i:x_{u,i}=+1} c_{i,k}^2}} \right), \quad \boldsymbol{\mu}_u^{(k)} = \frac{\mathbf{a}_u^{(k)}}{\|\mathbf{a}_u^{(k)}\|_2}, \quad \boldsymbol{\sigma}_u^{(k)} \leftarrow \sigma_0 \cdot \exp\left( -\frac{1}{2} \mathbf{b}_u^{(k)} \right). \quad (7)$$

The neural network $f_{\mathrm{nn}}(\cdot)$ captures nonlinearity, and is shared across the $K$ components. We normalize the mean, so as to be consistent with the use of cosine similarity which projects the representations onto a unit hypersphere. Note that $\sigma_0$ should be set to a small value, e.g., around $0.1$, since the learned representations are now normalized.

### 2.4 User-Controllable Recommendation

The controllability enabled by the disentangled representations can bring a new paradigm for recommendation. It allows a user to interactively search for items that are similar to an initial item except for some controlled aspects, or to explicitly adjust the disentangeld representation of his/her preference, learned by the system from his/her past behaviors, to actually match the current preference. Here we formalize the task of user-controllable recommendation, and illustrate a possible solution.

**Task definition**    Let $\mathbf{h}_* \in \mathbb{R}^d$ be the representation to be altered, which can be initialized as either an item representation or a component of a user representation. The task is to gradually alter its $j^{\mathrm{th}}$ dimension $h_{*,j}$, while retrieving items whose representations are similar to the altered representation. This task is nontrivial, since usually no item will have exactly the same representation as the altered one, especially when we want the transition to be smooth, monotonic, and thus human-understandable.

**Solution**    Here we illustrate our approach to this task. We first probe the suitable range $(a, b)$ for $h_{*,j}$. Let us assume that prototype $k_*$ is the prototype closest to $\mathbf{h}_*$. The range $(a, b)$ is decided such that: prototype $k_*$ remains the prototype closest to $\mathbf{h}_*$ if and only if $h_{*,j} \in (a, b)$. We can decide each endpoint of the range using binary search. We then divide the range $(a, b)$ into $B$ subranges, $a = a_0 < a_1 < a_2 \ldots < a_B = b$. We ensure that the subranges contain roughly the same number of items from concept $k_*$ when dividing $(a, b)$ . Finally, we aim to retrieve $B$ items $\{i_t\}_{t=1}^B \in \{1, 2, \ldots, M\}^B$ that belong to concept $k_*$, each from one of the $B$ subranges, i.e., $h_{i_t, j} \in (a_{t-1}, a_t]$. We thus decide the $B$ items by maximizing $\sum_{1 \le t \le B} e^{\frac{\text{COSINE}(\mathbf{h}_{i_t, -j}, \mathbf{h}_{*, -j})}{\tau}} + \gamma \cdot \sum_{1 \le t < t' \le B} e^{\frac{\text{COSINE}(\mathbf{h}_{i_t, -j}, \mathbf{h}_{i_{t'}, -j})}{\tau}}$, where $\mathbf{h}_{i,-j} = [h_{i,1}; h_{i,2}; \ldots; h_{i,j-1}; h_{i,j+1}; \ldots; h_{i,d}] \in \mathbb{R}^{d-1}$ and $\gamma$ is a hyper-parameter. We approximately solve this maximization problem sequentially using beam search [36].

Intuitively, selecting items from the $B$ subranges ensures that the items change monotonously in terms of the $j^{\mathrm{th}}$ dimension. On the other hand, the first term in the maximization problem forces the retrieved items to be similar with the initial item in terms of the dimensions other than $j$, while the second term encourages any two retrieved items to be similar in terms of the dimensions other than $j$.

## 3   Empirical Results

### 3.1   Experimental Setup

**Datasets**    We conduct our experiments on five real-world datasets. Specifically, we use the large-scale Netflix Prize dataset [4], and three MovieLens datasets of different scales (i.e., ML-100k, ML-1M, and ML-20M) [16]. We follow MultVAE [32], and binarize these four datasets by keeping ratings of four or higher while only keeping users who have watched at least five movies. We additionally collect a dataset, named AliShop-7C [2], from Alibaba's e-commerce platform Taobao. AliShop-7C contains user-item interactions associated with items from seven categories, as well as item attributes such as titles and images. Every user in this dataset clicks items from at least two categories. The category labels are used for evaluation only, and not for training.

**Baselines**    We compare our approach with MultDAE [32] and $\beta$-MultVAE [32], the two state-of-the-art methods for collaborative filtering. In particular, $\beta$-MultVAE is similar to $\beta$-VAE [19], and has a hyper-parameter $\beta$ that controls the strength of disentanglement. However, $\beta$-MultVAE does not learn disentangled representations, because it requires $\beta \ll 1$ to perform well.

Table 1: Collaborative filtering. All methods are constrained to have around $2Md$ parameters, where $M$ is the number of items and $d$ is the dimension of each item representation. We set $d = 100$.

| | | Metrics | | |
|---|---|---|---|---|
| Dataset | Method | NDCG@100 | Recall@20 | Recall@50 |
| AliShop-7C | MultDAE | 0.23923 ($\pm$0.00380) | 0.15242 ($\pm$0.00305) | 0.24892 ($\pm$0.00391) |
| | $\beta$-MultVAE | 0.23875 ($\pm$0.00379) | 0.15040 ($\pm$0.00302) | 0.24589 ($\pm$0.00387) |
| | Ours | **0.29148** ($\pm$0.00380) | **0.18616** ($\pm$0.00317) | **0.30256** ($\pm$0.00397) |
| ML-100k | MultDAE | 0.24487 ($\pm$0.02738) | 0.23794 ($\pm$0.03605) | 0.32279 ($\pm$0.04070) |
| | $\beta$-MultVAE | 0.27484 ($\pm$0.02883) | 0.24838 ($\pm$0.03294) | 0.35270 ($\pm$0.03927) |
| | Ours | **0.28895** ($\pm$0.02739) | **0.30951** ($\pm$0.03808) | **0.41309** ($\pm$0.04503) |
| ML-1M | MultDAE | 0.40453 ($\pm$0.00799) | 0.34382 ($\pm$0.00961) | 0.46781 ($\pm$0.01032) |
| | $\beta$-MultVAE | 0.40555 ($\pm$0.00809) | 0.33960 ($\pm$0.00919) | 0.45825 ($\pm$0.01039) |
| | Ours | **0.42740** ($\pm$0.00789) | **0.36046** ($\pm$0.00947) | **0.49039** ($\pm$0.01029) |
| ML-20M | MultDAE | 0.41900 ($\pm$0.00209) | 0.39169 ($\pm$0.00271) | **0.53054** ($\pm$0.00285) |
| | $\beta$-MultVAE | 0.41113 ($\pm$0.00212) | 0.38263 ($\pm$0.00273) | 0.51975 ($\pm$0.00289) |
| | Ours | **0.42496** ($\pm$0.00212) | **0.39649** ($\pm$0.00271) | 0.52901 ($\pm$0.00284) |
| Netflix | MultDAE | 0.37450 ($\pm$0.00095) | 0.33982 ($\pm$0.00123) | 0.43247 ($\pm$0.00126) |
| | $\beta$-MultVAE | 0.36291 ($\pm$0.00094) | 0.32792 ($\pm$0.00122) | 0.41960 ($\pm$0.00125) |
| | Ours | **0.37987** ($\pm$0.00096) | **0.34587** ($\pm$0.00124) | **0.43478** ($\pm$0.00125) |

**Hyper-parameters**   We constrain the number of learnable parameters to be around $2Md$ for each method so as to ensure fair comparison, which is equivalent to using $d$-dimensional representations for the $M$ items. Note that all the methods under investigation use two sets of item representations, and we do not constrain the dimension of user representations since they are not parameters. We set $d = 100$ unless otherwise specified. We fix $\tau$ to 0.1. We tune the other hyper-parameters of both our approach's and our baselines' automatically using the TPE method [6] implemented by Hyeropt [5].

## 3.2 Recommendation Performance

We evaluate the performance of our approach on the task of collaborative filtering for implicit feedback datasets [21], one of the most common settings for recommendation. We follow the experiment protocol established by the previous work [32] strictly, and use the same preprocessing procedure as well as evaluation metrics. The results on the five datasets are listed in Table 1.

We observe that our approach outperforms the baselines significantly, especially on small, sparse datasets. The improvement is likely due to two desirable properties of our approach. Firstly, macro disentanglement not only allows us to accurately represent the diverse interests of a user using the different components, but also alleviates data sparsity by allowing a rarely visited item to borrow information from other items of the same category, which is the motivation behind many hierarchical methods [50, 38]. Secondly, as we will show in Section 3.4, the dimensions of the representations learned by our approach are highly disentangled, i.e., independent, thanks to the micro disentanglement regularizer, which leads to more robust performance.

## 3.3 Macro Disentanglement

We visualize the high-dimensional representations learned by our approach on AliShop-7C in order to qualitatively examine to which degree our approach can achieve macro disentanglement. Specifically, we set $K$ to seven, i.e., the number of ground-truth categories, when training our model. We visualize the item representations and the user representations together using t-SNE [40], where we treat the $K$ components of a user as $K$ individual points and keep only the two components that have the highest confidence levels. The confidence of component $k$ is defined as $\sum_{i:x_{u,i}>0} c_{i,k}$, where $c_{i,k}$ is the value inferred by our model, rather than the ground-truth. The results are shown in Figure 2.

**Interpretability**   Figure 2c, which shows the clusters inferred based on the prototypes, is rather similar to Figure 2d that shows the ground-truth categories, despite the fact that our model is trained

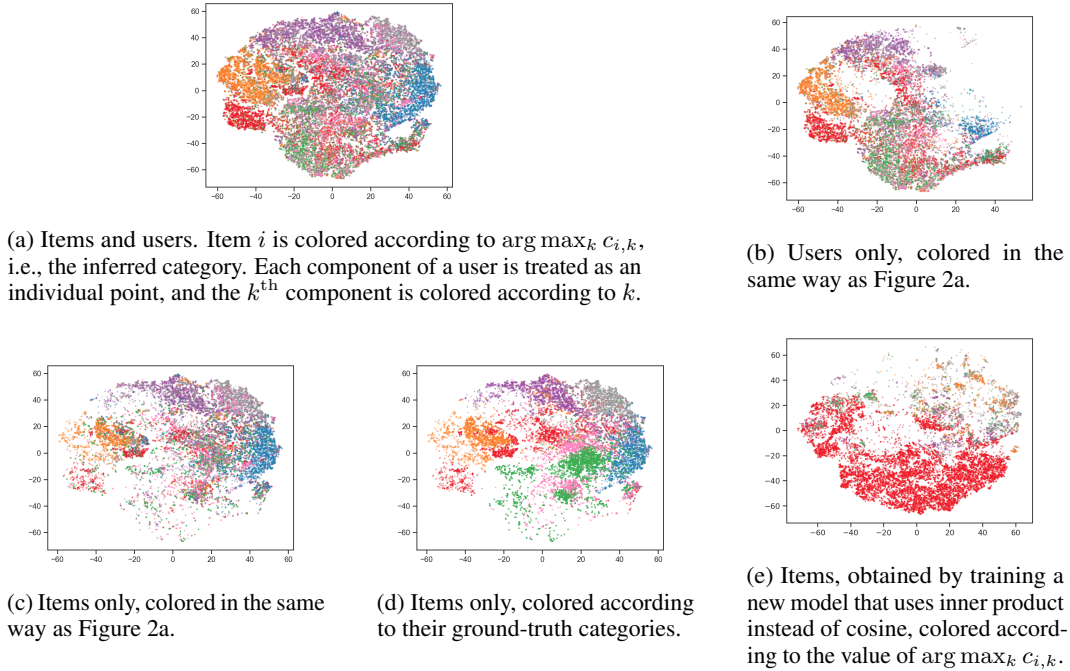

(a) Items and users. Item $i$ is colored according to $\arg\max_k c_{i,k}$, i.e., the inferred category. Each component of a user is treated as an individual point, and the $k^{\text{th}}$ component is colored according to $k$.

(b) Users only, colored in the same way as Figure 2a.

(c) Items only, colored in the same way as Figure 2a.

(d) Items only, colored according to their ground-truth categories.

(e) Items, obtained by training a new model that uses inner product instead of cosine, colored according to the value of $\arg\max_k c_{i,k}$.

Figure 2: The discovered clusters of items (see Figure 2c), learned unsupervisedly, align well with the ground-truth categories (see Figure 2d, where the color order is chosen such that the connections between the ground-truth categories and the learned clusters are easy to verify). Figure 2e highlights the importance of using cosine similarity, rather than inner product, to combat mode collapse.

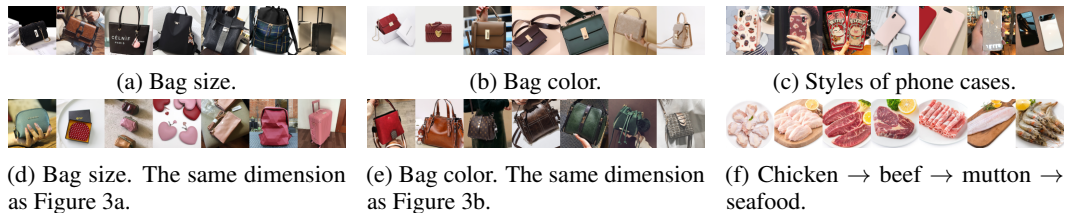

(a) Bag size.

(b) Bag color.

(c) Styles of phone cases.

(d) Bag size. The same dimension as Figure 3a.

(e) Bag color. The same dimension as Figure 3b.

(f) Chicken $\rightarrow$ beef $\rightarrow$ mutton $\rightarrow$ seafood.

Figure 3: Starting from an item representation, we gradually alter the value of a target dimension, and list the items that have representations similar to the altered representations (see Subsection 2.4).

without the ground-truth category labels. This demonstrates that our approach is able to discover and disentangle the macro structures underlying the user behavior data in an interpretable way. Moreover, the components of the user representations are near the correct cluster centers (see Figure 2a and Figure 2b), and are hence likely capturing the users' separate preferences for different categories.

**Cosine vs. inner product**   To highlight the necessity of using cosine similarity instead of the more commonly used inner product similarity, we additionally train a new model that uses inner product in place of cosine, and visualize the learned item representations in Figure 2e. With inner product, the majority of the items are assigned to the same prototype (see Figure 2e). In comparison, all seven prototypes learned by the cosine-based model are assigned a significant number of items (see Figure 2c). This finding supports our claim that a proper metric space, such as the one implied by the cosine similarity, is important for preventing mode collapse.

### 3.4   Micro Disentanglement

**Independence**   We vary the hyper-parameters related with micro disentanglement ($\beta$ and $\sigma_0$ for our approach, $\beta$ for $\beta$-MultVAE), and plot in Figure 4 the relationship between the level of independence achieved and the recommendation performance. Each method is evaluated with 2,000 randomly

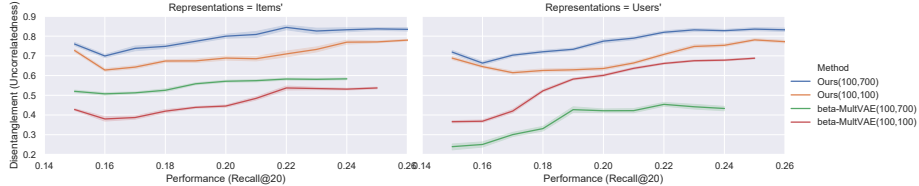

Figure 4: Micro disentanglement vs. recommendation performance. $(d, d')$ indicates $d$-dimensional item representations and $d'$-dimensional user representations. Note that $d' = Kd$. We observe that (1) our approach outperforms the baselines in terms of both performance and micro disentanglement, and (2) macro disentanglement benefits micro disentanglement, as $K = 7$ is better than $K = 1$.

sampled configurations on ML-100k. We quantify the level of independence achieved by a set of $d$-dimensional representations using $1 - \frac{2}{d(d-1)} \sum_{1 \leq i < j \leq d} |\text{corr}_{i,j}|$, where $\text{corr}_{i,j}$ is the correlation between dimension $i$ and $j$. Figure 4 suggests that high performance is in general associated with a relatively high level of independence. And our approach achieves a much higher level of independence than $\beta$-MultVAE. In addition, the improvement brought by using $K = 7$ instead of $K = 1$ reveals that macro disentanglement can possibly help improve micro disentanglement.

**Interpretability**   We train our model with $K = 7$, $d = 10$, $\beta = 50$ and $\sigma_0 = 0.3$, on AliShop-7C, and investigate the interpretability of the dimensions using the approach illustrated in Subsection 2.4. In Figure 3, we list some representative dimensions that have human-understandable semantics. These examples suggest that our approach has the potential to give users fine-grained control over targeted aspects of the recommendation lists. However, we note that not all dimensions are human-understandable. In addition, as pointed out by Locatello et al. [34], well-trained interpretable models can only be reliably identified with the help of external knowledge, e.g., item attributes. We thus encourage future efforts to focus more on (semi-)supervised methods [35].

## 4   Related Work

**Learning representations from user behavior**   Learning from user behavior has been a central task of recommender systems since the advent of collaborative filtering [43, 42, 46, 12, 21]. Early attempts apply matrix factorization [29, 45], while the more recent deep learning methods [49, 32, 31, 52, 11, 18] achieve massive improvement by learning highly informative representations. The entanglement of the latent factors behind user behavior, however, is mostly neglected by the black-box representation learning process adopted by the majority of the existing methods. To the extent of our knowledge, we are the first to study disentangled representation learning on user behavior data.

**Disentangled representation learning**   Disentangled representation learning aims to identify and disentangle the underlying explanatory factors [3]. $\beta$-VAE [19] demonstrates that disentanglement can emerge once the KL divergence term in the VAE [27] objective is aggressively penalized. Later approaches separate the information bottleneck term [48, 47] and the total correlation term, and achieve a greater level of disentanglement [9, 25, 8]. Though a few existing approaches [14, 10, 7, 13, 24] do notice that a dataset can contain samples from different concepts, i.e., follow a mixture distribution, their settings are fundamentally different from ours. To be specific, these existing approaches assume that each instance is from a concept, while we assume that each instance interacts with objects from different concepts. The majority of the existing efforts are from the field of computer vision [28, 15, 20, 30, 53, 14, 10, 39, 19]. Disentangled representation learning on relational data, such as graph-structured data, was not explored until recently [37]. This work focus on disentangling user behavior, another kind of relational data commonly seen in recommender systems.

## 5   Conclusions

In this paper, we studied the problem of learning disentangled representations from user behavior, and presented our approach that performs disentanglement at both a macro and a micro level. An interesting direction for future research is to explore novel applications that can be enabled by the interpretability and controllability brought by the disentangled representations.

**Acknowledgments**

The authors from Tsinghua University are supported in part by National Program on Key Basic Research Project (No. 2015CB352300), National Key Research and Development Project (No. 2018AAA0102004), National Natural Science Foundation of China (No. 61772304, No. 61521002, No. 61531006, No. U1611461), Beijing Academy of Artificial Intelligence (BAAI), and the Young Elite Scientist Sponsorship Program by CAST. All opinions, findings, and conclusions in this paper are those of the authors and do not necessarily reflect the views of the funding agencies.

## Footnotes

*Equal contribution. Work done at Alibaba.

[2]The dataset and our code are at `https://jianxinma.github.io/disentangle-recsys.html`.

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
