[Supplementary Material · alibaba-disentangle-recsys-supp.pdf]

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

,\mathbf{C}|\mathbf{x}_u)}\left[\ln \frac{q_{\boldsymbol{\theta}}(\mathbf{z}_u,\mathbf{C} \mid \mathbf{x}_u)}{p_{\boldsymbol{\theta}}(\mathbf{z}_u,\mathbf{C} \mid \mathbf{x}_u)}\right] + \mathbb{E}_{q_{\boldsymbol{\theta}}(\mathbf{z}_u,\mathbf{C}|\mathbf{x}_u)}\left[\ln \frac{p_{\boldsymbol{\theta}}(\mathbf{x}_u,\mathbf{z}_u,,\mathbf{C})}{q_{\boldsymbol{\theta}}(\mathbf{z}_u,\mathbf{C} \mid \mathbf{x}_u)}\right]\\
&= \mathbb{E}_{q_{\boldsymbol{\theta}}(\mathbf{z}_u,\mathbf{C}|\mathbf{x}_u)}\left[\ln \frac{q_{\boldsymbol{\theta}}(\mathbf{z}_u,\mathbf{C} \mid \mathbf{x}_u)}{p_{\boldsymbol{\theta}}(\mathbf{z}_u,\mathbf{C} \mid \mathbf{x}_u)}\right]\\
&\quad + \mathbb{E}_{q_{\boldsymbol{\theta}}(\mathbf{z}_u,\mathbf{C}|\mathbf{x}_u)}\left[\ln p_{\boldsymbol{\theta}}(\mathbf{x}_u \mid \mathbf{z}_u,\mathbf{C})\right] + \mathbb{E}_{q_{\boldsymbol{\theta}}(\mathbf{z}_u,\mathbf{C}|\mathbf{x}_u)}\left[\ln \frac{p_{\boldsymbol{\theta}}(\mathbf{z}_u,\mathbf{C})}{q_{\boldsymbol{\theta}}(\mathbf{z}_u,\mathbf{C} \mid \mathbf{x}_u)}\right]\\
&= D_{\mathrm{KL}}(q_{\boldsymbol{\theta}}(\mathbf{z}_u,\mathbf{C} \mid \mathbf{x}_u)\|p_{\boldsymbol{\theta}}(\mathbf{z}_u,\mathbf{C} \mid \mathbf{x}_u))\\
&\quad + \mathbb{E}_{q_{\boldsymbol{\theta}}(\mathbf{z}_u,\mathbf{C}|\mathbf{x}_u)}\left[\ln p_{\boldsymbol{\theta}}(\mathbf{x}_u \mid \mathbf{z}_u,\mathbf{C})\right] - D_{\mathrm{KL}}(q_{\boldsymbol{\theta}}(\mathbf{z}_u,\mathbf{C} \mid \mathbf{x}_u)\|p_{\boldsymbol{\theta}}(\mathbf{z}_u,\mathbf{C}))\\
&\geq \mathbb{E}_{q_{\boldsymbol{\theta}}(\mathbf{z}_u,\mathbf{C}|\mathbf{x}_u)}\left[\ln p_{\boldsymbol{\theta}}(\mathbf{x}_u \mid \mathbf{z}_u,\mathbf{C})\right] - D_{\mathrm{KL}}(q_{\boldsymbol{\theta}}(\mathbf{z}_u,\mathbf{C} \mid \mathbf{x}_u)\|p_{\boldsymbol{\theta}}(\mathbf{z}_u,\mathbf{C}))\\
&= \mathbb{E}_{p_{\boldsymbol{\theta}}(\mathbf{C})}\left[\mathbb{E}_{q_{\boldsymbol{\theta}}(\mathbf{z}_u|\mathbf{x}_u,\mathbf{C})}[\ln p_{\boldsymbol{\theta}}(\mathbf{x}_u \mid \mathbf{z}_u,\mathbf{C})] - D_{\mathrm{KL}}(q_{\boldsymbol{\theta}}(\mathbf{z}_u \mid \mathbf{x}_u,\mathbf{C})\|p_{\boldsymbol{\theta}}(\mathbf{z}_u))\right].
\end{aligned}
$$

Note that in the last line above, we have used

$$
\begin{aligned}
&D_{\mathrm{KL}}(q_{\boldsymbol{\theta}}(\mathbf{z}_u,\mathbf{C} \mid \mathbf{x}_u)\|p_{\boldsymbol{\theta}}(\mathbf{z}_u,\mathbf{C}))\\
&= D_{\mathrm{KL}}(q_{\boldsymbol{\theta}}(\mathbf{z}_u \mid \mathbf{x}_u,\mathbf{C})p_{\boldsymbol{\theta}}(\mathbf{C})\|p_{\boldsymbol{\theta}}(\mathbf{z}_u)p_{\boldsymbol{\theta}}(\mathbf{C}))\\
&= \mathbb{E}_{p_{\boldsymbol{\theta}}(\mathbf{C})}\left[D_{\mathrm{KL}}(q_{\boldsymbol{\theta}}(\mathbf{z}_u \mid \mathbf{x}_u,\mathbf{C})\|p_{\boldsymbol{\theta}}(\mathbf{z}_u))\right].
\end{aligned}
$$

$\square$

**Information bottleneck (IB) and total correlation (TC)**

$$\mathbb{E}_{p_{\mathrm{data}}(\mathbf{x}_u)}\left[D_{\mathrm{KL}}(q_{\boldsymbol{\theta}}(\mathbf{z}_u \mid \mathbf{x}_u,\mathbf{C})\|p_{\boldsymbol{\theta}}(\mathbf{z}_u))\right] = I_q(\mathbf{x}_u;\mathbf{z}_u) + D_{\mathrm{KL}}(q_{\boldsymbol{\theta}}(\mathbf{z}_u \mid \mathbf{C})\|p_{\boldsymbol{\theta}}(\mathbf{z}_u)).$$

*Proof.*

$$
\begin{aligned}
&\mathbb{E}_{p_{\mathrm{data}}(\mathbf{x}_u)}\left[D_{\mathrm{KL}}(q_{\boldsymbol{\theta}}(\mathbf{z}_u \mid \mathbf{x}_u,\mathbf{C})\|p_{\boldsymbol{\theta}}(\mathbf{z}_u))\right]\\
&= \mathbb{E}_{p_{\mathrm{data}}(\mathbf{x}_u)}\left[\mathbb{E}_{q_{\boldsymbol{\theta}}(\mathbf{z}_u|\mathbf{x}_u,\mathbf{C})}\left[\ln \frac{q_{\boldsymbol{\theta}}(\mathbf{z}_u \mid \mathbf{x}_u,\mathbf{C})}{p_{\boldsymbol{\theta}}(\mathbf{z}_u)}\right]\right]\\
&= \mathbb{E}_{p_{\mathrm{data}}(\mathbf{x}_u)}\left[\mathbb{E}_{q_{\boldsymbol{\theta}}(\mathbf{z}_u|\mathbf{x}_u,\mathbf{C})}\left[\ln \frac{q_{\boldsymbol{\theta}}(\mathbf{z}_u \mid \mathbf{x}_u,\mathbf{C})}{q_{\boldsymbol{\theta}}(\mathbf{z}_u \mid \mathbf{C})}\frac{q_{\boldsymbol{\theta}}(\mathbf{z}_u \mid \mathbf{C})}{p_{\boldsymbol{\theta}}(\mathbf{z}_u)}\right]\right]\\
&= \mathbb{E}_{p_{\mathrm{data}}(\mathbf{x}_u)}\left[\mathbb{E}_{q_{\boldsymbol{\theta}}(\mathbf{z}_u|\mathbf{x}_u,\mathbf{C})}\left[\ln \frac{q_{\boldsymbol{\theta}}(\mathbf{z}_u \mid \mathbf{x}_u,\mathbf{C})}{q_{\boldsymbol{\theta}}(\mathbf{z}_u \mid \mathbf{C})} + \ln \frac{q_{\boldsymbol{\theta}}(\mathbf{z}_u \mid \mathbf{C})}{p_{\boldsymbol{\theta}}(\mathbf{z}_u)}\right]\right]\\
&= \mathbb{E}_{p_{\mathrm{data}}(\mathbf{x}_u)}\left[D_{\mathrm{KL}}(q_{\boldsymbol{\theta}}(\mathbf{z}_u \mid \mathbf{x}_u,\mathbf{C})\|q_{\boldsymbol{\theta}}(\mathbf{z}_u \mid \mathbf{C}))\right] + \mathbb{E}_{q_{\boldsymbol{\theta}}(\mathbf{z}_u|\mathbf{x}_u,\mathbf{C})p_{\mathrm{data}}(\mathbf{x}_u)}\left[\ln \frac{q_{\boldsymbol{\theta}}(\mathbf{z}_u \mid \mathbf{C})}{p_{\boldsymbol{\theta}}(\mathbf{z}_u)}\right]\\
&= I_q(\mathbf{x}_u;\mathbf{z}_u) + \mathbb{E}_{q_{\boldsymbol{\theta}}(\mathbf{z}_u|\mathbf{C})}\left[\ln \frac{q_{\boldsymbol{\theta}}(\mathbf{z}_u \mid \mathbf{C})}{p_{\boldsymbol{\theta}}(\mathbf{z}_u)}\right]\\
&= I_q(\mathbf{x}_u;\mathbf{z}_u) + D_{\mathrm{KL}}(q_{\boldsymbol{\theta}}(\mathbf{z}_u \mid \mathbf{C})\|p_{\boldsymbol{\theta}}(\mathbf{z}_u)).
\end{aligned}
$$

Note that $p_{\mathrm{data}}(\mathbf{x}_u \mid \mathbf{C}) = p_{\mathrm{data}}(\mathbf{x}_u)$, and the mutual information $I_q(\mathbf{x}_u;\mathbf{z}_u)$ is under the joint distribution $q_{\boldsymbol{\theta}}(\mathbf{z}_u,\mathbf{x}_u \mid \mathbf{C}) = q_{\boldsymbol{\theta}}(\mathbf{z}_u \mid \mathbf{x}_u,\mathbf{C})p_{\mathrm{data}}(\mathbf{x}_u \mid \mathbf{C}) = q_{\boldsymbol{\theta}}(\mathbf{z}_u \mid \mathbf{x}_u,\mathbf{C})p_{\mathrm{data}}(\mathbf{x}_u)$. $\square$

Table 2: Dataset statistics.

| | AliShop-7C | ML-100k | ML-1M | ML-20M | Netflix |
|---|---|---|---|---|---|
| # of users | 10,668 | 603 | 6,038 | 136,677 | 463,435 |
| # of items | 20,591 | 569,7 | 3,605 | 20,108 | 17,769 |
| # of interactions | 767,493 | 47,922 | 836,452 | 9,990,030 | 56,880,037 |
| # of held-out users | 4,000 | 50 | 500 | 10,000 | 40,000 |

## A.2 Experimental Details

**Datasets**  Datasets are preprocessed using the script provided by $\beta$-MultVAE. Half of the held-out users are used for validation, while the other half of the held-out users are for testing.

**Infrastructure**  We implement our model with Tensorflow, and conduct our experiments with:

- CPU: Intel(R) Xeon(R) CPU E5-2699 v4 @ 2.20GHz.
- RAM: DDR4 1TB.
- GPU: 8x GeForce GTX 1080 Ti.
- Operating system: Ubuntu 18.04 LTS.
- Software: Python 3.6; NumPy 1.15.4; SciPy 1.2.0; scikit-learn 0.20.0; TensorFlow 1.12.

**Hyper-parameter search**  We treat $K$ as a hyper-parameter to be tuned and do not directly set $K$ to the ground truth when evaluating its performance on recommendation tasks, so as to ensure a fair comparison with the baselines. We set $d = 100$. We fix $\tau$ to 0.1. The neural network $f_{\mathrm{nn}}(\cdot)$ in our model is a multilayer perceptron (MLP), whose input and output are constrained to be $d$-dimensional and $2d$-dimensional, respectively. We use the tanh activation function. We apply dropout before every layers, except the last layer. The model is trained using Adam. We then tune the other hyper-parameters of both our approach's and our baselines' automatically using the TPE method implemented by Hyepropt. We let Hyperopt conduct 200 trials to search for the optimal hyper-parameter configuration for each method on the validation of each dataset. The hyper-parameter search space is specified as follows:

- The standard deviation of the prior $\sigma_0 \in [0.075, 0.5]$.
- The strength of micro disentanglement $\beta \in [0, 100]$.
- The number of macro factors $K \in \{1, 2, 3, \ldots, 20\}$.
- The learning rate $\in [10^{-8}, 1]$.
- L2 regularization $\in [10^{-12}, 1]$.
- Dropout rate $\in [0.05, 1]$.
- The number of hidden layers in a neural network $\in \{0, 1, 2, 3\}$.
- The number of neurons in a hidden layer $\in \{50, 100, 150, \ldots, 700\}$.

**The number of macro factors**  Our initial implementation adaptively adjusts the number of macro factors $K$ during training. To be specific, we set $K$ as a sufficiently large value at the beginning and shrink its value after every training epoch if the Jensen–Shannon (JS) divergence between $\{p_{i|k}\}_{i=1}^{M}$ and $\{p_{i|k'}\}_{i=1}^{M}$ for some $k \neq k'$ is negligible compared to a predefined threshold, where $p_{i|k} := p_{\boldsymbol{\theta}}(c_{i,k} = 1) / \sum_{i'} p_{\boldsymbol{\theta}}(c_{i',k} = 1)$. We, however, do not find this adaptive strategy to be significantly better than the naïve strategy that treats $K$ as a hyper-parameter to be tuned by Hyperopt, since the adaptive strategy introduces extra computational cost as well as a new hyper-parameter.

## A.3 Implementation Details

See Algorithm 1.

**Algorithm 1** The training procedure. We add $10^{-8}$ to prevent division-by-zero wherever appropriate.

1: **input:** $\mathbf{x}_u = \{x_{u,i} : \text{user } u \text{ clicks item } i, \text{ i.e., } x_{u,i} = 1\}$.
2: **parameters:** Concept prototypes $\mathbf{m}_k \in \mathbb{R}^d$ for $k = 1, 2, \ldots, K$; Item representations $\mathbf{h}_i \in \mathbb{R}^d$ for $i = 1, 2, \ldots, M$; Context representations $\mathbf{t}_i \in \mathbb{R}^d$ for $i = 1, 2, \ldots, M$; Parameters of a neural network $f_{\mathrm{nn}} : \mathbb{R}^d \to \mathbb{R}^{2d}$.         $\triangleright$ All these parameters are referred to collectively as $\boldsymbol{\theta}$.
3: **function** PROTOTYPECLUSTERING
4:     **for** $i = 1, 2, \ldots, M$ **do**
5:         $s_{i,k} \leftarrow \mathbf{h}_i^\top \mathbf{m}_k / (\tau \cdot \|\mathbf{h}_i\|_2 \cdot \|\mathbf{m}_k\|_2), \quad k = 1, 2, \ldots, K.$
6:         $\mathbf{c}_i \sim \text{GUMBEL-SOFTMAX}([s_{i,1}; s_{i,2}; \ldots; s_{i,K}]).$         $\triangleright$ At test time, $\mathbf{c}_i$ is set to the mode.
7:     **return** $\{\mathbf{c}_i\}_{i=1}^M$
8: **function** ENCODER($\mathbf{x}_u, \{\mathbf{c}_i\}_{i=1}^M$)
9:     **for** $k = 1, 2, \ldots, K$ **do**
10:         $(\mathbf{a}_k, \mathbf{b}_k) \leftarrow f_{\mathrm{nn}} \left( \dfrac{\sum_{i : x_{u,i} = +1} c_{i,k} \cdot \mathbf{t}_i}{\sqrt{\sum_{i : x_{u,i} = +1} c_{i,k}^2}} \right), \; \boldsymbol{\mu}^{(k)} \leftarrow \mathbf{a}_k / \|\mathbf{a}_k\|_2, \; \boldsymbol{\sigma}^{(k)} \leftarrow \sigma_0 \cdot \exp\left(-\tfrac{1}{2}\mathbf{b}_k\right).$
11:     $\boldsymbol{\mu}_u \leftarrow [\boldsymbol{\mu}^{(1)}; \boldsymbol{\mu}^{(2)}; \ldots; \boldsymbol{\mu}^{(K)}], \quad \boldsymbol{\sigma}_u \leftarrow [\boldsymbol{\sigma}^{(1)}; \boldsymbol{\sigma}^{(2)}; \ldots; \boldsymbol{\sigma}^{(K)}], \quad \boldsymbol{\epsilon} \sim \mathcal{N}(\mathbf{0}, \mathbf{I}).$
12:     $\mathbf{z}_u = \boldsymbol{\mu}_u + \boldsymbol{\epsilon} \circ \boldsymbol{\sigma}_u.$ $\triangleright$ $\mathbf{z}_u$ is set to $\boldsymbol{\mu}_u$ at test time. "$\circ$" stands for element-wise multiplication.
13:     **return** $\mathbf{z}_u, D_{KL}(\mathcal{N}(\boldsymbol{\mu}_u, \mathrm{diag}(\boldsymbol{\sigma}_u)) \| \mathcal{N}(\mathbf{0}, \sigma_0 \cdot \mathbf{I}))$
14: **function** DECODER($\mathbf{z}_u, \{\mathbf{c}_i\}_{i=1}^M$)
15:     $p_{u,i} \leftarrow \sum_{k=1}^K c_{i,k} \cdot \exp(\mathbf{z}_u^{(k)^\top} \mathbf{h}_i / (\tau \cdot \|\mathbf{z}_u^{(k)}\|_2 \cdot \|\mathbf{h}_i\|_2)), \quad i = 1, 2, \ldots, M.$
16:     $[p_{u,1}; p_{u,2}; \ldots; p_{u,M}] \leftarrow \text{SOFTMAX}([\ln p_{u,1}; \ln p_{u,2}; \ldots; \ln p_{u,M}]).$
17:     $\triangleright$ We replace the SOFTMAX($\cdot$) above with SAMPLED-SOFTMAX($\cdot$), and compute $p_{u,i}$ only if $x_{u,i} = 1$ or item $i$ is sampled, when $M$ is very large.
18:     **return** $\{p_{u,i}\}_{i=1}^M$
19: $\{\mathbf{c}_i\}_{i=1}^M \leftarrow$ PROTOTYPECLUSTERING( ).
20: $\mathbf{z}_u, D_{\mathrm{KL}} \leftarrow$ ENCODER($\mathbf{x}_u, \{\mathbf{c}_i\}_{i=1}^M$).
21: $\{p_{u,i}\}_{i=1}^M \leftarrow$ DECODER($\mathbf{z}_u, \{\mathbf{c}_i\}_{i=1}^M$).
22: $L = -\beta \cdot D_{\mathrm{KL}} + \sum_{i : x_{u,i} = 1} \ln p_{u,i}.$
23: $\boldsymbol{\theta} \leftarrow$ Update $\boldsymbol{\theta}$ to maximize $L$, using the gradient $\nabla_{\boldsymbol{\theta}} L$.