[Reviews · NeurIPS 2019]

Reviewer 1



This paper propose a VAE-based model to learn disentangled representations based on user behavior, including both macro one and micro one. They model user representation as latent variable z, item representation as condition c, and user-item interactions as x. The whole paper is well-organized and well written. It is very easy to follow and understand. They compare MacridVAE with state of art collaborative filtering approaches, and demonstrate superiority of their method. But, we can find too many assumptions in the paper, like independence and sufficient assumptions. It would be great if the authors can provide more insights about why such assumptions important. Since one item may fall into more than one categories. Hierarchical recommendation algorithms are not new to the community, it is not clear how such disentangled representation differ from them, and what the superiority of using disentangled representations. It seems that disentangled representation is just a two-level representation. Another confusing thing is the setting of parameter K. Is it always set to ground truth value? What if we do not know the ground truth value? I have read authors' response, and I tend to increase the score.

Reviewer 2



Originality: - The task is not new. The proposed approach is in line with some of the prior art, but the specific approach is new. Related work is fairly cited. Quality: - the technique appears to be technically sound. - claims are well supported by theoretical analysis and experimental results - this is a complete piece of work. - Choosing the number of components K: May be I missed this, but I think the paper does not discuss how to choose K, what happens if the chosen K is way off from the actual K. Figure 2 and Figure 3 are with K = 7 which it appears is the number of ground truth categories for that dataset. line 209: we do not constrain the dimension of user representations since they are not parameters. I am not sure I understand this / agree with this, especially when comparing different methods. Clarity: - the paper is well written and well organized. Significance: - The contributions of the paper are relevant and significant. - the authors will also be releasing a new recommendation ratings dataset. I have carefully considered the authors' response. The rebuttal looks fair. But my question was more like a clarification. So, it doesn't change my scores.

Reviewer 3



- The macro disentanglement resembles a cluster assignment process, and the micro disentaglement (encourage independence between dimensions) is a ordinary method for learning disentangled representation. However, the whole framework makes sense to me, and the use of Gumbel-softmax trick and cosine similarity is also reasonable. - It'd be better to show visualizations of baselines (e.g. MultDAE) in Figure 2, so that we can see the comparison. As learning such an item representation (distinguished by category, like clustering) is not hard. The micro disentanglement (Figure 3) is interesting, but the quantitative measurement is missing. - I'd like to see more experimental analysis, like ablation study of the macro and micro disentanglement (e.g. set K=1 to remove macro disentanglement). - Is there a reason to account for the superior performance, especially on sparse data? Maybe the proposed macro-micro structure alleviates the data sparsity problem in some way? It might be nitpicking that line 218 says "consistently outperforms baselines" which is not exactly true. - The main concern I have is the lack of baselines, as it only compares with two methods from a recent work[30], but there are many CF baselines like BPR-MF are missing, and they often show competitive performance. --- The rebuttal addressed most of my concerns, hence I decided to raise my score.

[Author Response · NeurIPS 2019]

We thank the reviewers for the valuable suggestions and appreciate the positive feedback.

# 1 To both Reviewer #1 & Reviewer #2:

## 1.1 How is the number of components $K$ decided? What if $K$ differs from the ground truth?

We do not directly set $K$ to the ground truth when evaluating its performance on recommendation tasks (Table 1), so as to ensure a fair comparison with the baselines. Instead, we set $K$ as a sufficiently large value initially and shrink its value during training if the JS divergence between $\{p_{i|k}\}_{i=1}^{M}$ and $\{p_{i|k'}\}_{i=1}^{M}$ for some $k \neq k'$ is negligible compared to a predefined threshold, where $p_{i|k} := p_\theta(c_i = k)/\sum_{i'} p_\theta(c_{i'} = k)$. As for Figure 2 & 3, which are about interpretability, we set $K$ to the ground truth, i.e., $K = 7$. We will revise the paper to ensure these details are included.

We experiment with $K \in \{1, 2, \ldots, 20\}$ on dataset Shop-7C and have the following observations. (1) When $K$ is much smaller than the ground-truth value, the performance of our approach on recommendation tasks degrades and becomes close to or even slightly below that of the baselines. The quality of the micro disentanglement also suffers in the sense that the dimensions become more correlated and less interpretable. Note that Figure 4 from Subsection 3.4 also shows that the learned representations are much less micro-disentangled when $K = 1$ is used in place of $K = 7$. (2) On the other hand, when $K$ is larger than the ground truth, the performance rarely improves, and the degradation is not as severe when it happens. We will update the supplemental material to include the relevant results.

# 2 To Reviewer #1

## 2.1 On how our work is fundamentally different from hierarchical recommender systems.

The traditional hierarchical methods usually cluster items and/or users to abstract representations at a higher level of granularity, while our disentangled approach can further decompose an item as well as a user's preference according to the micro factors (e.g., size, color, and price) to obtain fine-grained interpretable representations. The latter direction is a largely unexplored topic in the literature of recommender systems that may enable potential novel applications such as user-controllable recommendation. We adopt the hierarchical design mainly to accommodate the fact that different categories (or macro factors) can be associated with distinctly different sets of micro factors (see Q&A 2.2 below).

## 2.2 On the relevance of the independence and sufficiency assumptions in real applications.

Actually, it is the real-world application of recommender systems that motivates us to assume the independence and sufficiency. (1) Industrial recommender systems involve highly diverse items, and a user's preference is closely connected to the product categories. A user's preference regarding battery life is applicable to laptops but not to lipsticks. Even for a common property such as price, the user may simultaneously prefer low-priced laptops and high-priced lipsticks. Such complicated user preference can only be effectively discovered if the assumptions of independence and sufficiency at the macro level are imposed. (2) On the other hand, encouraging independence between the dimensions, which is a common idea behind many disentangled representation learning methods, benefits interpretability, because it facilitates the discovery of semantically independent micro factors such as sizes, colors, and prices, which are valuable for explainable recommendation. (3) We can indeed relax the assumptions by instead using a hierarchical prior such as the normal-inverse-Wishart prior. However, we value scalability more than the marginal improvement.

# 3 To Reviewer #3

## 3.1 Whether the reported baselines are enough.

Our work focuses more on interpretability and controllability rather than performance. Admittedly, more baselines will necessarily make the results more convincing. However, we must note that the baseline [30] that we do compare with is the state-of-the-art. The recent results of arXiv:1907.06902 show that Liang's method [30] (1) is the strongest neural approach when it comes to top-n recommendation tasks and (2) outperforms or is at least on par with the strongest non-neural ranking methods such as SLIM. Note that SLIM is typically stronger than BPR variants such as BPR-MF and BPR-kNN. Nevertheless, we are willing to include more baselines such as SLIM and BPRs in the revised version.

## 3.2 Ablation studies of the macro and micro disentanglement, e.g., by setting $K = 1$.

These results can be found in Subsection 3.4 (see Figure 4). Figure 4 shows that (1) $K = 7$ leads to a higher level of micro disentanglement than $K = 1$, which indicates the necessity of macro disentanglement, and that (2) strengthening micro disentanglement often brings better performance. We will revise the paper to make these results more visible.

## 3.3 Quantitative measurement of the micro disentanglement.

We have quantitatively measured the statistical independence of the dimensions in Subsection 3.4. The results (see Figure 4) show that our approach significantly improves upon the baselines when it comes to micro disentanglement.

## 3.4 Is there a reason to account for the superior performance, especially on sparse data?

The macro-micro design alleviates data sparsity by allowing a rarely visited item to borrow information from other items of the same category, which is the motivation behind many hierarchical methods [49]. On the other hand, sparse data tend to affect the robustness and stability of an algorithm, and Bengio et al. [3] suggest that disentangled representations are more robust since prediction with such representations tends to be stable when a few nuisance factors vary.

## 3.5 Whether we plan to release the source code.

Yes, we plan to release the source code, along with all the datasets used, once the paper is publicly published.

**Reference** [49] Zhang & Koren. Efficient Bayesian hierarchical user modeling for recommendation systems. SIGIR'07.

[Meta-Review · NeurIPS 2019]

The reviewers all felt positively about the paper, though the scores were somewhat borderline, meaning this paper was right on the threshold for inclusion in the conference. A discussion was initiated to discuss the authors' rebuttal, which resulted in some positive movement of reviewer scores. Overall reviewers felt that the rebuttal seemed reasonable: most of the issues are clarifications, that the authors do a good job of clarifying. A few are hard to address in the rebuttal, such as the inclusion of additional baselines. Overall the reviewers are positive regarding the release of new datasets, and the significance of the results, though the reviewers broadly agreed that additional experimental comparisons would make this paper more convincing.